# COVID-19 Vaccination in Pediatrics: Was It Valuable and Successful?

**DOI:** 10.3390/vaccines11020214

**Published:** 2023-01-18

**Authors:** Mohamed Ahmed Raslan, Sara Ahmed Raslan, Eslam Mansour Shehata, Amr Saad Mahmoud, Nagwa A. Sabri, Khalid J. Alzahrani, Fuad M. Alzahrani, Saleh Alshammeri, Vasco Azevedo, Kenneth Lundstrom, Debmalya Barh

**Affiliations:** 1Drug Research Centre, Cairo P.O. Box 11799, Egypt; 2Department of Obstetrics and Gynecology, Faculty of Medicine, Ain Shams University, Cairo P.O. Box 11566, Egypt; 3Department of Clinical Pharmacy, Faculty of Pharmacy, Ain Shams University, Cairo P.O. Box 11566, Egypt; 4Department of Clinical Laboratories Sciences, College of Applied Medical Sciences, Taif University, P.O. Box 11099, Taif 21944, Saudi Arabia; 5Department of Optometry, College of Applied Medical Sciences, Qassim University, P.O. Box 6688, Buraydah 51542, Saudi Arabia; 6Department of Genetics, Ecology and Evolution, Institute of Biological Sciences, Federal University of Minas Gerais, Belo Horizonte 31270-901, Brazil; 7PanTherapeutics, Route de Lavaux 49, CH1095 Lutry, Switzerland; 8Institute of Integrative Omics and Applied Biotechnology (IIOAB), Nonakuri, Purba Medinipur 721172, India

**Keywords:** COVID-19, vaccination, types, pediatrics, adverse events, efficacy, safety

## Abstract

Background: The mass vaccination of children against coronavirus 2019 disease (COVID-19) has been frequently debated. The risk–benefit assessment of COVID-19 vaccination versus infection in children has also been debated. Aim: This systematic review looked for answers to the question “was the vaccination of our children valuable and successful?”. Methods: The search strategy of different articles in the literature was based on medical subject headings. Screening and selection were based on inclusion/exclusion criteria. Results and Discussion: The search results revealed that the majority of the reported adverse events after COVID-19 vaccination in pediatrics were mild to moderate, with few being severe. Injection site discomfort, fever, headache, cough, lethargy, and muscular aches and pains were the most prevalent side effects. Few clinical studies recorded significant side effects, although the majority of these adverse events had nothing to do with vaccination. In terms of efficacy, COVID-19 disease protection was achieved in 90–95% of cases for mRNA vaccines, in 50–80% of cases for inactivated vaccines, and in 58–92% of cases for adenoviral-based vaccines in children and adolescents. Conclusions: Based on available data, COVID-19 immunizations appear to be safe for children and adolescents. Furthermore, multiple studies have proven that different types of vaccines can provide excellent protection against COVID-19 in pediatric populations. The efficacy of vaccines against new SARS-CoV-2 variants and the reduction in vaccine-related long-term adverse events are crucial for risk–benefit and cost-effectiveness assessments; therefore, additional safety studies are required to confirm the long-term safety and effectiveness of vaccinations in children.

## 1. Introduction

Severe acute respiratory syndrome coronavirus 2 (SARS-CoV-2) was discovered at the end of 2019 in Wuhan, China. Since then, SARS-CoV-2 has spread throughout the world, resulting in the COVID-19 pandemic [1].The COVID-19 pandemic has claimed a total of more than 650 million cases and more than 6.6 million deaths worldwide as of 2 January 2023.

Vaccination is an effective strategy for controlling viral diseases, as it can contribute to lowering the risk of severe disease and fatality, limiting disease transmission, and potentially developing herd immunity. A variety of COVID-19 vaccines have been developed and widely used in adults, and some have also received conditional approval for mass vaccination in children in some countries. In China, for example, inactivated viral vaccines from the Beijing Institute of Biological Products, the Wuhan Institute of Biological Products, and Sinovac Biotech have received provisional authorization for the immunization of children aged 3 to 11 years [2].

On the other hand, the BNT162b2 mRNA vaccine, co-developed by BioNTech and Pfizer, and the mRNA-1273 vaccine developed by Moderna have been approved by the FDA for use in children from the age of 6 months to 5 years in the US. Furthermore, the FDA also authorized the use of the protein subunit-based vaccine NVX-CoV2373, developed by Novavax Inc., USA, in children aged 12 to 17 years in December 2021 [3,4].

Despite the fact that approximately 26.6 million COVID-19 vaccine doses had been administered to individuals aged 5 to 17 years in the US as of February 2022, the question of whether COVID-19 vaccination should be recommended for children and adolescents remains debatable. Generally, children and adolescents are excluded from clinical trials due to their vulnerable age; published studies on safety and efficacy in this group are, therefore, limited [5]. However, the case is different for COVID-19. A quick PubMed search immediately identified 11 publications on COVID-19 vaccines and children, most importantly a review by Zimmermann et al. (2022) [6].

Because of the low incidence of severe, acute COVID-19 in children and the potential harms of vaccinations and SARS-CoV-2 infection, the risk–benefit assessment of immunization in this age group is more complicated. One of the most significant reasons to vaccinate healthy children and adolescents is to reduce the incidence of long-term side effects and complications resulting from SARS-CoV-2 infection. Furthermore, the aims of COVID-19 vaccinations are to decrease viral transmission, provide adequate vaccination supply, reduce healthcare costs, and avoid quarantine, school closures, and other lockdown measures [6].

## 2. Materials and Methods

The review aimed to systematically study the safety and efficacy of COVID-19 vaccines in children and adolescents and evaluate potential adverse events associated with vaccinations. We are not disputing the enormous advantages of mass COVID-19 vaccinations, which undoubtedly have significantly decreased COVID-19 morbidity and mortality. Medical Subject Headings (MeSH) was used in the search strategy. The MeSH terms used included pediatrics, vaccinations, COVID-19, safety, and immunological studies to systematically search the PubMed and MEDLINE databases.

The PubMed databases were searched to collect all types of manuscripts (e.g., reviews, research articles, short communications, book chapters, case reports, etc.) dealing with the safety and efficacy of COVID-19 vaccines in pediatrics and written in the English language until November 2022. Steps including title and abstract screening, full-text review, and duplicate removal were used to screen the studies for inclusion and exclusion criteria.

Our inclusion criteria primarily focused on published literature that assessed the safety and efficacy of COVID-19 vaccines in pediatrics. The studies should have been conducted on males or females belonging to the age group ranging from 6 months to 19 years. Furthermore, comparison of vaccine efficacy and risk–benefit assessment in children and adolescents versus adults was also included.

A total of 6564 articles about COVID-19 vaccinations were found in PubMed. (Figure 1) The articles were screened for inclusion and exclusion criteria, and from the search results, we included 34 articles in this systematic review.

## 3. Results and Discussion

### 3.1. Disease Incidence, Symptoms, and Complications

Children and adolescents tend to develop milder viral infection than adults, but all ages are vulnerable to infection and serious complications [7]. COVID-19 is a respiratory disease that appears in children as flu-like sickness with fever and cough [8] (Figure 2).

Approximately one-third of the children and adolescents requiring hospitalization were admitted to intensive care [9]. Children with a history of respiratory or cardiovascular diseases are likely to be more vulnerable to infection with SARS-CoV-2. A recent study highlighted an increased vulnerability to infection in individuals with asthma or renal impairment, but more data are needed to confirm these findings [10].

Furthermore, a newly characterized disease known as multisystem inflammatory syndrome in children (MIS-C) or pediatric inflammatory multisystem syndrome (PIMS-TS) has been found in a subgroup of pediatric patients soon after SARS-CoV-2 infection. Studies describing MIS-C/PIMS-TS are rapidly appearing. Comorbid diseases and conditions have been reported in a few multicenter studies, with rates ranging from 3% to 25%, with wide variation [11]. Obesity and cardiovascular disease were the most common comorbid conditions. Other, far less frequent, co-occurring neuromuscular, oncologic, immunosuppressive, autoimmune, and hereditary disorders have also been reported [12].

### 3.2. Different Types of Vaccines

According to the WHO statistics provided on 23 November 2022, 175 vaccine candidates have been authorized for clinical trials, with another 199 in preclinical testing [13] (Figure 3). As of today, a total of thirteen vaccines, including inactivated vaccines, viral vector vaccines, mRNA vaccines, and protein subunit vaccines, have received Emergency Use Authorization (EUA) by the WHO [14].

### 3.3. Post-Vaccination Adverse Reactions

Although the present COVID-19 vaccines have received EUA and have proven to be safe in clinical studies, they have shown different adverse events, including fever, headache, tiredness, injection site irritation, and nausea (Figure 4). Complications emerged in certain participants as mass vaccinations took place, leading to some deaths of patients with cardiovascular disorders such as arteriosclerosis [15]. Additionally, cardiac arrest occurred in one subject in a Phase III trial for the BNT162b2 vaccine, although it was not considered to be associated with the vaccination [16]. Potential complications caused by COVID-19 vaccinations fall into the following six categories: heart diseases (such as myocarditis), coagulation disorders (such as thrombocytopenia), immune diseases (such as allergic reactions, autoimmune hepatitis, and autoimmune thyroid diseases), lymphatic system diseases, nervous system diseases (such as functional neurological disorders), and other diseases (such as Rowell’s syndrome, macular rash, and chilblain-like lesions) [15]. Despite the rare occurrence of serious adverse events, the association between vaccines and these disorders needs to be investigated [15].

### 3.4. Were COVID-19 Vaccinations Useful and Successful in Pediatrics?

Different studies on SARS-CoV-2 vaccines in children and adolescents have shown that these vaccines appear to be effective and safe [17] (Table 1). Furthermore, the results have indicated similar or better immune responses in children/adolescents than in adults. As a result, an increasing number of nations have started to vaccinate pediatric populations. Vaccines for children aged 12 and above have recently been registered in the US, Canada, and the EU.

Furthermore, the Canadian Pediatric Society, the American Academy of Pediatrics, and the Advisory Committee on Immunization Practices have urged for vaccination of all children and adolescents over the age of 12. However, the UK has delayed this decision owing to lack of convincing data concerning the safety and necessity of SARS-CoV-2 immunization in children [18]. Currently, EU countries have recommended COVID-19 vaccination of adolescents from 12 to 17 years old. On the other hand, 10 countries from the EU recommended booster doses for children below 18 years of age [19]. 

The obvious benefit of immunizing children and teenagers is protection against COVID-19. Although the symptoms are usually moderate or minimal in children, severe incidences have been reported [20].

Furthermore, as with adults, there is some indication that a significant percentage of children who catch the virus have at least one symptom that lasts longer than four months. These post-COVID-19 symptoms may impede children’s everyday activities and development. Furthermore, COVID-19 may cause some permanent health implications [21]. In this context, preliminary findings indicate that COVID-19 may cause long-term symptoms such as tiredness, muscle and joint pain, sleeplessness, breathing issues, and palpitations lasting up to six months. The symptoms are difficult to differentiate from other conditions often found in children and adolescents [22].

Other benefits of vaccination in children and adolescents may include alleviation of back-to-school concerns. For over a year, the mandatory home confinement of children has resulted in the accumulation of developmental, educational, and psychological issues [23]. On the other hand, providing protection against COVID-19 would also stimulate the reintroduction of routine pediatric care, including immunization against other infectious diseases, which was significantly disrupted during the pandemic and lockdown periods [24]. Furthermore, the demand for caregivers for sick children at home would diminish.

Because children and adolescents account for more than one-fourth of the overall population, herd immunity to control COVID-19 spreading and mitigating severe conditions cannot be achieved without extensive immunization of pediatric populations. Children account for 14.3% of people diagnosed with COVID-19 in the US. Although clinical symptoms of COVID-19 in children are often mild or individuals are asymptomatic compared with adults, a small number of individuals develop severe symptoms, necessitating hospitalization, and this may even lead to death [25]. 

Recent observations in the US revealed a rise in morbidity and severe cases among children and adolescents, which is of great concern. Furthermore, acute respiratory infections are the most prevalent diseases in children, and COVID-19 symptoms in children are difficult to differentiate from other respiratory diseases. As a source of infection, infected children may play a large role in community transmission due to their interaction within families, in nurseries, and in schools. Children thus represent a significant demographic group that needs COVID-19 vaccines [25].

**Table 1 vaccines-11-00214-t001:** Clinical trials on safety and efficacy of COVID-19 vaccines in children and adolescents.

Intervention/ Control Groups	Age Range	Follow-Up Duration	Number of Doses and Schedule	Vaccine Name	Vaccine Type	Phase or Study Type	Location	Clinical Outcome	Ref.
1131 (Intervention) 1129 (Control)	12–15 years	4.7 months	2 doses Days 0 and 21	BNT162b2	mRNA vaccine	III	USA	Recipients showed a favorable safety profile and produced stronger immune responses than young adults. The vaccine was highly effective against COVID-19.	[26]
1565 (Intervention) 751 (Control)	5–11 years	2.3 months	2 doses Days 0 and 21	BNT162b2	mRNA vaccine	I/II/III	USA, Spain, Finland, and Poland	The vaccine was safe, immunogenic, and efficacious.	[27]
2489 (Intervention) 1243 (Control)	12–17 years	4.6 months	2 doses Days 0 and 28	mRNA-1273	mRNA vaccine	II/III	USA	The vaccine showed an acceptable safety profile in adolescents and efficiently prevented COVID-19	[28]
436 (Intervention) 114 (Control)	3–17 years	4.1 months	2 doses Days 0 and 28	CoronaVac	Inactivated vaccine	I/II	China	The vaccine was well tolerated and safe and induced humoral responses in children and adolescents.	[29]
755 (Intervention) 252 (Control)	3–17 years	Not reported	3 doses Days 0, 28, and 56	BBIBP-CorV	Inactivated vaccine	I/II	China	The vaccine was safe and well tolerated and elicited robust humoral responses against SARS-CoV-2 infection after two doses.	[30]
100 (Intervention) 50 (Control)	6–17 years	Not reported	2 doses Days 0 and 56	Ad5-nCoV	Adenovirus vaccine	IIb	China	A single vaccine dose was safe and induced robust immune responses in children and adolescents. The 56-day booster dose provided limited effect.	[31]
448 (Intervention) 487 (Control)	12–17 years	11 months	3 doses Days 0, 28, and 56	ZyCov-D	DNA vaccine	III	India	The vaccine was safe, immunogenic, and efficacious.	[32]
468 (Intervention) 156 (Control)	5–18 years	6 months	2 doses Days 0 and 28	CORBEVAX™	Protein Sub-unit vaccine	II/III	India	The safety profile of the vaccine in children and adolescents was good. Both humoral and cellular immune responses were comparable to those found in adults.	[33]
118 (Intervention)	12–18 years	6 months	2 doses Days 0, 21, or 42	BNT162b2	mRNA vaccine	II	Thailand	Healthy adolescents had good immune responses to the fractional dose regimen of BNT162b2.	[34]
2969 (Intervention) 864 (Control)	6–11 years	82 days	2 doses Days 0 and 28	mRNA-1273	mRNA vaccine	II/III	USA, Canada	Two 50 μg doses were safe and effective in inducing immune responses and preventing COVID-19.	[35]
963 (Intervention)	3–17 years	Short-term follow-up	2 doses Days 0 and 28	CoronaVac	Inactivated vaccine	III	Chile	The vaccine was safe and immunogenic against SARS-CoV-2 and its variants. Neutralizing antibodies were identified against the Delta and Omicron variants.	[36]
39,422 (Test-positive cases) 140,690 (Test-negative controls)	12–19 years	Short-term follow-up	2 doses (mRNA vaccine) or 1 dose (Adenovirus vaccine)	BNT162b2, mRNA-1273, and Ad26.COV2.S	mRNA vaccines and Adenovirus vaccine	Test-negative case control	USA	Slightly better protection against the COVID-19 Delta variant than in adults. Booster doses were recommended to enhance time-related mitigated immunization.	[37]
1364 children and adolescents	5–15 years	Short-term follow-up	2 doses or 1 dose	BNT162b2	mRNA vaccine	Prospective cohort study	USA	Two doses of the BNT162b2 vaccine were effective in preventing both asymptomatic and symptomatic SARS-CoV-2 Omicron infection.	[38]

Since vaccination in children and adolescents is important, the WHO has granted EUA of seven COVID-19 vaccines in pediatric populations [39]. China has granted EUA of inactivated vaccines in children aged 3 to 17 years [40]. The Centers for Disease Control and Prevention (CDC) has also approved emergency vaccination in individuals aged 6 months to 17 years (Table 2) [3].

### 3.5. What Are the Risks of COVID-19 Vaccinations in Children and Adolescents?

Numerous non-life-threatening adverse events related to COVID-19 vaccines have been documented in the pediatric age group. However, these side effects are infrequent, with a reported adverse event frequency of less than 0.2% [48]. The majority of reported adverse events for different vaccine types were minor and temporary. Frequent adverse effects may include pain at the injection site, tiredness, fever, chills, dizziness, headache, crying, and loss of appetite. Furthermore, seizures, stroke, myocarditis, pericarditis, MIS-C, hematuria, chest pain, menstruation disturbance, appendicitis, behavioral and otologic adverse events, and others have been documented [48,49,50,51,52,53,54,55,56].

In a case series that included a group of five adolescents aged 15 to 17 years with obesity/overweight (BMI 24.8–30), the individuals showed characteristic myocarditis symptoms after the first or second dose of the BNT162b2 vaccine. A considerable rise in troponin serum concentration was detected, followed by a rapid decrease within 3 to 4 days. COVID-19 vaccine-induced myocarditis appears to be a benign condition with rapid clinical recovery, although total resolution of the inflammatory process may take more than three months. Further studies and follow-ups are necessary to establish the long-term effects of COVID-19 vaccine-induced myocarditis [49].

### 3.6. Comorbidities and Vaccination Hazards

#### 3.6.1. Children Suffering from the following Allergic Diseases

Children with allergic rhinitis, conjunctivitis, atopic dermatitis, and food allergies can be vaccinated in stable disease status, which is defined as no disease exacerbation and stable condition for at least 3 months under standard therapeutic treatment [25]. Furthermore, individuals who are allergic to dust mites, pollen, alcohol, cefotaxime, and penicillin can also be vaccinated in stable status [25].

#### 3.6.2. Children with Asthma

Bronchial asthma does not exclude COVID-19 immunization. In the remission stage of asthma, vaccination should be administered (including inhaled corticosteroids). Vaccination should be avoided during acute asthma episodes, especially when glucocorticoids are taken systemically. COVID-19 vaccination can be given alongside anti-IgE monoclonal antibody treatment and allergen-specific immunotherapy, but not on the same day [56]. Patients are advised to select vaccination on days when no immunotherapy is scheduled.

#### 3.6.3. Children with Impaired Immune Function

Children with congenital or acquired decreased immune function can be vaccinated with the inactivated vaccine, as the same safety standards are met as for immunocompetent children [25]. However, the strength and duration of immunological protection in children with impaired immune function will most likely be inferior due to their weaker immune responses. Furthermore, according to CDC recommendations, mRNA vaccinations can be given to immunocompromised children aged 6 months to 17 years [57].

#### 3.6.4. Children Who Have Been Previously Diagnosed with COVID-19

After 6 months of infection, one dose of COVID-19 vaccine can be given [58].

#### 3.6.5. Children with Cardiovascular Disorders

The incidence of cardiovascular complications (e.g., myocarditis or pericarditis) is low in children (i.e., pooled rate of 37.76 per million) [59] after COVID-19 vaccinations. Due to lack of data on the effect of vaccination on children with cardiovascular disorders, it is advisable to evaluate risks versus benefits before starting vaccinations.

#### 3.6.6. Children with Renal Disorders

There are limited data available on the effect of COVID-19 vaccinations on both adult and pediatric patients with chronic kidney diseases. 

However, it has been shown that SARS-CoV-2 infections may cause some rare renal adverse reactions, such as IgA nephropathy, acute interstitial nephritis, antineutrophil cytoplasmic autoantibody vasculitis, and tubulointerstitial nephritis [60,61]. It has been recommended that caution should be taken in vaccinating children with renal disorders.

#### 3.6.7. Children with Diabetes

COVID-19 immunization was shown to be safe and did not cause any significant changes in glycemic control in adolescents and young adults with Type 1 diabetes (T1DM) [62].

### 3.7. Children and Vaccine Booster Recommendations

The FDA has further granted EUA of booster vaccinations with the BNT162b2 vaccine in children aged 5–11 years [63]. An intramuscular booster dose of 10 mcg (same as the primary doses) may be given 5 months after completion of two primary doses, or in moderately or severely immunocompromised children, a three-dose primary series.

In a case-control test-negative design, the vaccine effectiveness (VE) of BNT162b2 (Pfizer-BioNTech) in preventing COVID-19 hospitalization was evaluated. The results during Delta variant predominance indicated that in adolescents aged 16–17, vaccine effectiveness 14 to 149 days after the second dose was 76%, while after more than 150 days, VE after the second dose was 46%. On the other hand, VE increased to 86% after more than 7 days from the third booster dose. However, during Omicron variant predominance, the third booster dose restored VE to 81% [64]. 

A national cohort study in Singapore assessed the incidence of confirmed SARS-CoV-2 infections and hospitalization among 249,763 adolescents aged 12–17 years vaccinated with BNT162b2 during the Delta (B.1.617.2), and Omicron (B.1.1.529) variant wave. The results showed that two vaccine doses compared with no vaccination provided VE of 66% against the Delta variant infection, 25% against the Omicron infection, 83% against the Delta variant-associated hospitalization, and 75% against the Omicron variant-associated hospitalization. On the other hand, the third (booster) vaccine dose provided vaccine effectiveness of 56% against the Omicron variant infection and 94% against the Omicron-associated hospitalization [65].

### 3.8. Readiness of Parents to Vaccinate Themselves and Their Children

Vaccine reluctance and refusal has become more common in those who are younger, poorly educated, less healthy, and have doubts about the efficacy and safety of vaccines. A cross-sectional study in Puyang city, China, was conducted to investigate the hesitancy and willingness of parents to vaccinate themselves and their children with a COVID-19 vaccine booster dose. The results showed that 95.4% and 95.0% of participants who had completed two primary doses but did not match the booster criteria were willing to get a booster vaccine dose for themselves and their children, respectively. On the other hand, 40.3% of those who matched the booster dose requirements were vaccine hesitant [66].

Another study in Taizhou, China, showed that 41.8% of parents were undecided on whether to give their children COVID-19 vaccine boosters or not [67]. This means that better health advising, awareness, and education are required.

## 4. Conclusions

From available accumulated data, it seems that COVID-19 vaccinations are safe for children and adolescents, and numerous studies have confirmed that different types of vaccines can provide excellent protection against COVID-19 in children and adolescents. On the other hand, current information has so far suggested potential incidences of vaccine-related adverse events, some of which may be severe, although no deaths have been reported. Since COVID-19 vaccines became available, their EUA has been granted in children and adolescents in different corners of the world to prevent the further spread of the pandemic. However, as safety is of utmost importance, it is necessary to conduct well-planned and -executed clinical trials on vaccine safety and efficacy in children and adolescents. Precautions should also be taken to minimize the occurrence of adverse events, especially in children with pre-existing health issues.

## Figures and Tables

**Figure 1 vaccines-11-00214-f001:**
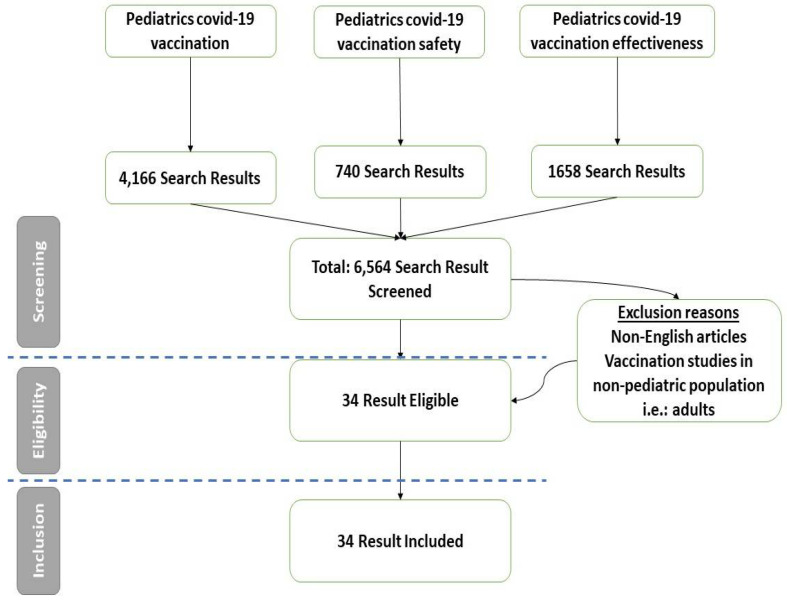
Flowchart of the article selection process.

**Figure 2 vaccines-11-00214-f002:**
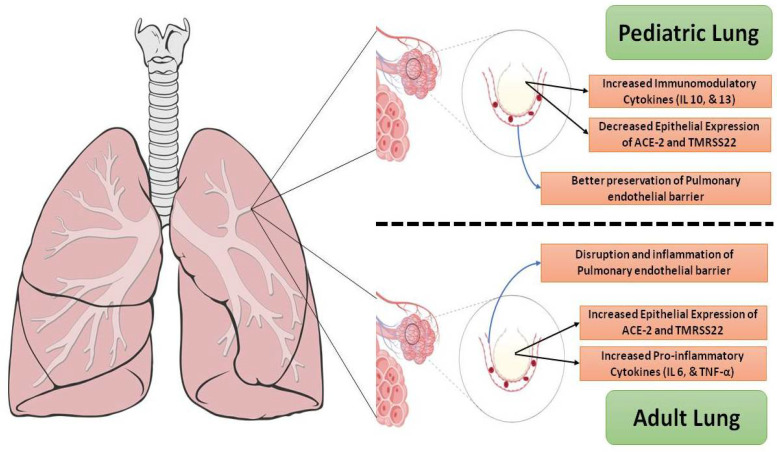
Mechanisms influencing the vulnerability of adults and children to SARS-CoV-2 infection. Increased expression of mediators essential for viral entry into airway epithelial cells (ACE-2 and TMPRSS2) in adults combined with the proinflammatory milieu may predispose the adult lung to serious pulmonary injury and progression to acute respiratory distress syndrome (ARDS). The pediatric lung has greater expression of immunomodulatory cytokines and possibly a decreased expression of viral entry mediators.

**Figure 3 vaccines-11-00214-f003:**
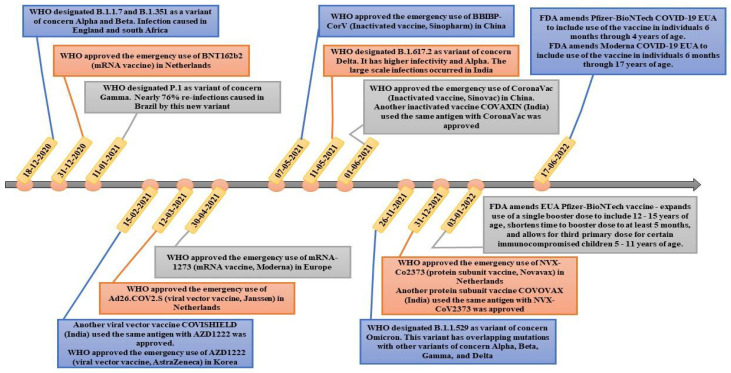
A chronology of significant events in the development of COVID-19 vaccines. The Emergency Use Authorization (EUA) of BNT162b2 happened shortly after the appearance of alpha and beta variants in the UK in December 2020. The WHO granted EUA for the adenovirus vector-based vaccine AZD1222, the inactivated vaccines by Sinopharm and Sinovac, and the protein subunit vaccine NVX-Co2373 in 2021. The FDA approved the use of the mRNA-1273 vaccine in children and adolescents from 6 months to 17 years of age in 2022.

**Figure 4 vaccines-11-00214-f004:**
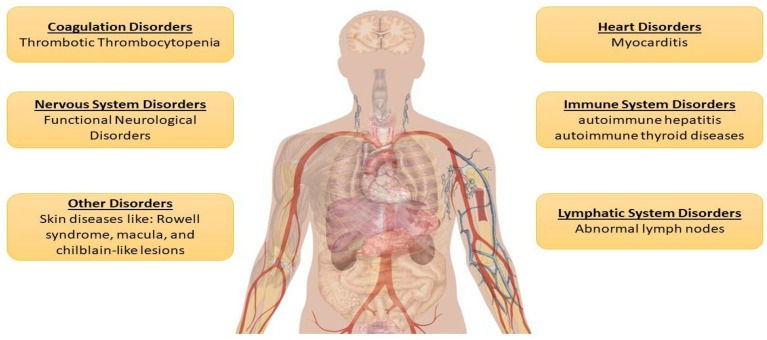
Possible COVID-19 vaccine complications. The following complications have been potentially associated with COVID-19 vaccines: (1) coagulation disorders (thrombocytopenia); (2) heart diseases (myocarditis); (3) lymphatic system diseases; (4) immune diseases (allergic reactions, autoimmune hepatitis, and thyroid diseases); (5) nervous system disorders (functional neurological disorders); and (6) other diseases (Rowell’s syndrome, macular rash, and chilblain-like lesions)].

**Table 2 vaccines-11-00214-t002:** List of COVID-19 vaccines granted EUA by the WHO, the FDA, and different health agencies [3,4,25,29,41,42,43,44,45,46,47].

Vaccine Name/ Manufacturer	Vaccine Type	Date of Approval	Age Group	Protective Efficacy
BNT162b2 (Pfizer/BioNTech, USA/Germany)	mRNA vaccine	14 January 2021	6 months to 4 years	95%
mRNA-1273 (Moderna, USA)		3 February 2021	6 months to 5 years	94.1%
Covishield (Serum institute of India, India)	Adenoviral vector vaccine	1 March 2021	18 years and older	63.09%
AZ1222 (AstraZeneca/University of Oxford, UK)			18 years and older	63.09%
Ad26.COV2.S (Johnson & Johnson, USA)		17 March 2021	18 years and older	66.9%
CONVIDECIA (Ad5-nCoV-S) (CanSino Biologics Inc., China)		19 May 2022	18 years and older	58 to 92%
BBIBP-CorV (CNBG, China)	Inactivated vaccine	7 May 2021	3 to 12 years	78.1%
CoronaVac (Sinovac, China)		1 June 2021	3 to 17 years	50.7%
COVAXIN (Bharat Biotech International Ltd., India)		3 November 2021	2 to 18 years	68 to 93%
Novavax (NVX-CoV2373) (Novavax Inc., USA)	Protein subunit	20 December 2021	12 to 17 years	80%
COVOVAX (Serum institute of India, India)		17 December 2021	12 to 17 years	86.3 to 89.7%

## Data Availability

Not applicable.

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
