# Peer review of "COVID-19 Vaccination in Pediatrics: Was It Valuable and Successful?"

_vaccines, 2023, doi:10.3390/vaccines11020214_

Round 1

Reviewer 1 Report

I recommended the publication of the manuscript in the current version

Author Response

Respected Reviewer

We would like to express our thanks and gratitude for your great effort in reviewing the manuscript which are highly appreciated.  

Kind Regards

Reviewer 2 Report

As the authors of this manuscript indicate, the utility and wisdom of vaccinating young children against SARS-CoV-2 has been controversial. The objective of this manuscript is to review clinical trials and other datasets to inform the question of safety and efficacy of such vaccines in pediatric populations.  The authors represent institutions in a broad range of countries, which is viewed positively.

I have to major criticisms of this manuscript:

1. It seems to be excessive long and would benefit substantially by shortening. Description of the question at hand (vaccines in children) does not start until page 9 and preceded by a lengthy description of the different types of COVID vaccines that are provided in numerous other reviews (Section 3.2).  I would suggest deleting or greatly shortening those sections describing the different vaccines (including elimination of Table 1) and perhaps the section on pathogenesis, which is also well reviewed by many others.  Table 2 is valuable and should be retained.

2. The fundamental conclusion presented in the abstract (more study is required and there have been a limited number of studies [there are alwaysw a limited number of studies]) is vague and essentially meaningless.  In contrast, a perfect conclusion is stated in lines 368-370 - I suggest that latter conclusion be copied to the abstract.

Additional comments:

 o Title: consider changing to "Was it valuable and successful?"

 o Line 31: suggest chaning to "The efficacy of mRNA vaccines was approximately 95% in terms of [what - preventing hospitalizations, clinical disaese, ?]

 o The statistics on cumulative death percentage described on lines 48-52 seem erroneous and do not reflect the information in reference [2].  To me, you are suggesting that for the US, the cummulative mortality rate is 43% (to my knowledge only about 1% of confirmed cases resulted in death).  The statement "is this South and/or North American" seems to be a left over from manuscript writing and should be resolved.  Finally, I believe the number of cases and cumulative mortality have been significantly higher in Europe than the Americas, but I am not sure of that because I'm not sure what "Americas" means as written.

Overall, my opinion is that there is valuable information in this literature review but that it needs to be shortened and cleaned up as described above.

Author Response

Respected Reviewer 2

Thank you for your valuable comments, notes and recommendations, which we have taken into account in the revised version. All changes are highlighted  in the revised manuscript.

As the authors of this manuscript indicate, the utility and wisdom of vaccinating young children against SARS-CoV-2 has been controversial. The objective of this manuscript is to review clinical trials and other datasets to inform the question of safety and efficacy of such vaccines in pediatric populations.  The authors represent institutions in a broad range of countries, which is viewed positively.

I have two major criticisms of this manuscript:

Comment #1

  1. It seems to be excessive long and would benefit substantially by shortening. Description of the question at hand (vaccines in children) does not start until page 9 and preceded by a lengthy description of the different types of COVID vaccines that are provided in numerous other reviews (Section 3.2).  I would suggest deleting or greatly shortening those sections describing the different vaccines (including elimination of Table 1) and perhaps the section on pathogenesis, which is also well reviewed by many others.  Table 2 is valuable and should be retained.

Reply

 We have taken into consideration your comments and have revised the manuscript accordingly.

----------------------------------------------------------------------------------------

Comment #2

The fundamental conclusion presented in the abstract (more study is required and there have been a limited number of studies [there are always a limited number of studies]) is vague and essentially meaningless.  In contrast, a perfect conclusion is stated in lines 368-370 - I suggest that latter conclusion be copied to the abstract.

Reply

We have followed your suggestion and implemented the text in the Abstract.

----------------------------------------------------------------------------------------

Comment #3

Title: consider changing to "Was it valuable and successful?"

Reply

We have changed the title as suggested.

----------------------------------------------------------------------------------------

Comment # 4

Line 31: suggest changing to "The efficacy of mRNA vaccines was approximately 95% in terms of [what - preventing hospitalizations, clinical disease?]

Reply

The text has been revised as suggested.

----------------------------------------------------------------------------------------

Comment #5

The statistics on cumulative death percentage described on lines 48-52 seem erroneous and do not reflect the information in reference [2].  To me, you are suggesting that for the US, the cumulative mortality rate is 43% (to my knowledge only about 1% of confirmed cases resulted in death).  The statement "is this South and/or North American" seems to be a left over from manuscript writing and should be resolved.  Finally, I believe the number of cases and cumulative mortality have been significantly higher in Europe than the Americas, but I am not sure of that because I'm not sure what "Americas" means as written.

Reply

As this information is not essential to the topic of the manuscript, we have deleted the section to reduce the length of manuscript.

---------------------------------------------------------------------------------------

Reviewer 3 Report

Line 31 – 33 - 

 In terms of mRNA vaccines 31 was the efficacy is approximately 95% effective in preventing COVID-19 among children and ado-32 lescents, 50-80% for inactivated vaccines, 63-7% for adenoviral-based vaccines, indicating high effi-33 cacy for this age group.

This sentence does not make sense

Line 37-38 – Needs to be rewritten – poor English usage

Line 49 (is this South and/or North America) – needs to be deleted and then the Americas needs to be defined

Line 52 – extraneous A

Line 67-70 – This text is extremely dated – must be updated

Line 80 should read severe, acute COVID-19 in children

Line 124 – Comorbid diseases and conditions

Line 230 – This COVID should be these covid

Liine 244 – Herd Immunity implies that the vaccine will prevent infection when in reality it is best at preventing severe disease

Line 262 – once again this is a dated statement, the US has recommended much more expanded access than just 12-17 years of age

Table 3 lists the List of COVID-19 vaccines granted EUA by the WHO in children.  This citation is over 15 months old and outdated.  This table should have the ages that the EUA has been extended to.

Line 273 – following which vaccine?

Section 3.12 – you bounce around between mRNA, inactivated and other vaccines.  I would suggest reorganizing to list adverse effects with one vaccine.

 3.13.1 -  What does in stable status mean?

3.13.3 – actually mRNA vaccines can be administered to people who are immunocompromised based on CDC recommendations

Lines 373-375 – confusing

375 – outmost - utmost

Author Response

Respected Reviewer 

Thank you for your valuable comments, notes and recommendations regarding the manuscript. We have taken into consideration all your suggestions and revised the manuscript accordingly.

Comment #1

Line 31 – 33 - In terms of mRNA vaccines was the efficacy is approximately 95% effective in preventing COVID-19 among children and adolescents, 50-80% for inactivated vaccines, 63-7% for adenoviral-based vaccines, indicating high efficacy for this age group. This sentence does not make sense

Reply

We have revised the text accordingly.

---------------------------------------------------------------------------------------

Comment #2

Line 37-38 – Needs to be rewritten – poor English usage

Reply

We have revised the text.

----------------------------------------------------------------------------------------

Comment #3

Line 49 (is this South and/or North America) – needs to be deleted and then the Americas needs to be defined

Reply

As this information is not essential to the topic of the manuscript, we have deleted the section to reduce the length of manuscript.

----------------------------------------------------------------------------------------

Comment #4

Line 52 – extraneous A

Reply

We have revised the text accordingly..

----------------------------------------------------------------------------------------

Comment #5

Line 67-70 – This text is extremely dated – must be updated

Reply

We have updated the text..

----------------------------------------------------------------------------------------

Comment #6

Line 80 should read severe, acute COVID-19 in children

Reply

We have added “severe” to the text.

----------------------------------------------------------------------------------------

Comment #7

Line 124 – Comorbid diseases and conditions

Reply

The suggested revision was made.

----------------------------------------------------------------------------------------

Comment #8

Line 230 – This COVID should be these covid

Reply

We have revised the text as “These COVID”.

----------------------------------------------------------------------------------------

Comment #9

Line 244 – Herd Immunity implies that the vaccine will prevent infection when in reality it is best at preventing severe disease

Reply

The suggested corrections were done.

----------------------------------------------------------------------------------------

Comment #10

Line 262 – once again this is a dated statement, the US has recommended much more expanded access than just 12-17 years of age

Reply

The text has been updated.

----------------------------------------------------------------------------------------

Comment #11

Table 3 lists the List of COVID-19 vaccines granted EUA by the WHO in children.  This citation is over 15 months old and outdated.  This table should have the ages that the EUA has been extended to.

Reply

As suggested Table 3 (now Table 2)  has been updated.

----------------------------------------------------------------------------------------

Comment #12

Line 273 – following which vaccine?

Reply

The BNTB162b2 vaccine is now mentioned.

----------------------------------------------------------------------------------------------------

Comment #13

Section 3.12 – you bounce around between mRNA, inactivated and other vaccines.  I would suggest reorganizing to list adverse effects with one vaccine.

Reply

The adverse have now been summarized.

----------------------------------------------------------------------------------------

Comment #14

3.13.1 - What does in stable status mean?

Reply

We have added the definition for stable status to the text.

----------------------------------------------------------------------------------------

Round 2

Reviewer 2 Report

Thank you for considering my suggestions.  I believe the manuscript has been substantially improved.  I do have one comment about the abstract: on lines 33 and 34, you state figures for efficacy of 95% (mRNA vaccines) and 63.7% (adenoviral vaccines) - those are incredibly specific.  Should you not make those figures ranges like you do for inactivated vaccines?

Author Response

Respected Reviewer 2

Thanks for your valuable comment and suggestion which were taken in consideration and performed as per recommended

Comment

Thank you for considering my suggestions.  I believe the manuscript has been substantially improved.  I do have one comment about the abstract: on lines 33 and 34, you state figures for efficacy of 95% (mRNA vaccines) and 63.7% (adenoviral vaccines) - those are incredibly specific.  Should you not make those figures ranges like you do for inactivated vaccines?

Reply

   Thanks for your valuable reviewing, suggestions and recommendations which are highly appreciated.

   Kindly, as per your valuable recommendation the figures ranges for the inactivated vaccines {figures for efficacy of 95% (mRNA vaccines) and 63.7% (adenoviral vaccines)} were removed from the abstract on lines 33 and 34 in the updated version of the manuscript.

Reviewer 3 Report

The only area that I do not see addressed in this review is that authors make not mention of booster recommendations for children, which is the stage of the pandemic for many countries at this time.

All other previous concerns have been addressed.

Author Response

Respected Reviewer 

Thanks for your great effort in reviewing the manuscript which are highly appreciated, kindly your valuable comment was addressed and added in the updated version of the manuscript.

Comment

The only area that I do not see addressed in this review is that authors make not mention of booster recommendations for children, which is the stage of the pandemic for many countries at this time.

All other previous concerns have been addressed.

Reply

Thanks for your effort and time for reviewing the manuscript in such a concise, informative and professional model.   

Kindly your valuable comment has been taken into consideration and a cited paragraph was added concerning the booster recommendations for children in the updated submitted version of the manuscript.